# Porcine Parvovirus Infection Causes Pig Placenta Tissue Damage Involving Nonstructural Protein 1 (NS1)-Induced Intrinsic ROS/Mitochondria-Mediated Apoptosis

**DOI:** 10.3390/v11040389

**Published:** 2019-04-25

**Authors:** Jianlou Zhang, Jinghui Fan, Yan Li, Shuang Liang, Shanshan Huo, Xing Wang, Yuzhu Zuo, Dan Cui, Wenyan Li, Zhenyu Zhong, Fei Zhong

**Affiliations:** 1Laboratory of Molecular Virology and Immunology, College of Animal Science and Technology/College of Veterinary Medicine, Hebei Agricultural University, Hebei Engineering and Technology Research Center of Veterinary Biotechnology, Baoding 071000, China; zhangjianlouauh@163.com (J.Z.); Jinghui76@163.com (J.F.); dyly@hebau.edu.cn (Y.L.); huoshanshan567@163.com (S.H.); wangxing4549@126.com (X.W.); 13653125036@163.com (Y.Z.); cuidanhcy@126.com (D.C.); 2Department of Immunology, University of Texas Southwestern Medical Center, Dallas, TX 75390-9093, USA; shuang.liang@utsouthwestern.edu; 3Department of Biology, School of Medicine, Hebei University, Baoding 071000, China; liwenyan358@sohu.com

**Keywords:** Porcine parvovirus, NS1 protein, apoptosis reactive oxygen species, mitochondria damage, intrinsic pathway

## Abstract

Porcine parvovirus (PPV) is an important pathogen causing reproductive failure in pigs. PPV-induced cell apoptosis has been recently identified as being involved in PPV-induced placental tissue damages resulting in reproductive failure. However, the molecular mechanism was not fully elucidated. Here we demonstrate that PPV nonstructural protein 1 (NS1) can induce host cell apoptosis and death, thereby indicating the NS1 may play a crucial role in PPV-induced placental tissue damages and reproductive failure. We have found that NS1-induced apoptosis was significantly inhibited by caspase 9 inhibitor, but not caspase 8 inhibitor, and transfection of NS1 gene into PK-15 cells significantly inhibited mitochondria-associated antiapoptotic molecules Bcl-2 and Mcl-1 expressions and enhanced proapoptotic molecules Bax, P21, and P53 expressions, suggesting that NS1-induced apoptosis is mainly through the mitochondria-mediated intrinsic apoptosis pathway. We also found that both PPV infection and NS1 vector transfection could cause host DNA damage resulting in cell cycle arrest at the G1 and G2 phases, trigger mitochondrial ROS accumulation resulting in mitochondria damage, and therefore, induce the host cell apoptosis. This study provides a molecular basis for elucidating PPV-induced cell apoptosis and reproductive failure.

## 1. Introduction

Porcine parvovirus (PPV) is one of the most important pathogens causing reproductive failure in sows, characterized by infertility, abortion, stillbirth, malformed fetus and mummification. It causes huge economic losses to the swine industry worldwide [1,2]. PPV is classified within the genus Parvovirus of the family Parvoviridae, with a single-stranded DNA genome of ~5 kb encoding three capsid proteins (VP1, VP2, and VP3) and three nonstructural proteins (NS1, NS2, and NS3) [3]. VP2 is the major component of the capsid proteins, possessing strong antigenicity to stimulate host immune responses [4,5] and mediating virus infection by interaction with the PPV receptor. VP1 is the minor component of the virus capsid, and is also involved in viral replication [6]. NS1 is an important nonstructural protein participating in viral genome replication, transcriptional regulation, and host cell pathogenicity [7]. NS2 has been identified to be involved in capsid assembly and nuclear export [8]. The functions of VP3 and NS3 are still unknown.

PPV-induced reproductive failure has been shown to be closely related to virus-mediated cell death and tissue damage [9]. Like other animal parvoviruses, PPV mainly infects the rapidly dividing cells in the placenta and fetus. PPV infection across the placental barrier can cause pregnant sow endometritis, epithelial cell death, and fetal infection, causing especially endothelial tissue damage [10]. PPV-induced cell death and tissue damage are considered to be an important pathogenic factor in sow reproductive failure. Therefore, investigation of the mechanism of PPV-induced cell death and tissue damage is essential for an understanding of PPV-induced reproductive failure. Based on the pathogenicity of other animal parvoviruses, which can induce host cell apoptosis and death [11,12,13], PPV was first postulated and then also identified to have the ability to induce host cell apoptosis [14,15], which was considered a crucial factor in PPV-induced cell death and tissue damage. However, the molecular mechanism underlying PPV-induced apoptosis remained unclear.

Investigations into the mechanism have shown that PPV-induced apoptosis mainly involves the ROS-induced intrinsic pathway. It has been shown that PPV can induce host apoptosis by activation of P53 and accumulation of ROS [14,15]. However, which PPV-encoded protein is mainly involved in PPV-induced apoptosis was not clear, and the more detailed molecular mechanism of PPV-induced apoptosis remained inadequately understood. In this study, we present evidence to indicate that the PPV-encoded NS1 contributes to PPV-induced apoptosis, which is mainly involved in the mitochondria-mediated intrinsic apoptosis pathway. 

## 2. Materials and Methods

### 2.1. Plasmid, Virus, Cells, and Tissue

The pcDNA3.1A plasmid used to construct eukaryotic expression vectors for PPV-encoded genes was purchased from Invitrogen (Carlsbad, CA, USA). A virulent PPV strain was a generous gift from Professor Weiquan Liu (College of Biology, China Agricultural University). Porcine kidney (PK)-15 cells used for PPV amplification and apoptosis assay were purchased from the Center for Type Culture Collection of China in Beijing (Beijing, China), cultured in Dulbecco’s Modified Eagle’s Medium (DMEM) supplemented with 10 % (*v*/*v*) fetal bovine serum (FBS), 100 U/mL penicillin, 100 μg/mL streptomycin, and 2 mM L-glutamine at 37 ℃ in a 5 % CO_2_ atmosphere incubator. The placental tissue collected from PPV-induced aborted and normal labored sows was provided by the local pig farms.

### 2.2. Reagents

Monoclonal antibodies against Bcl-2, Bax, P53, P21, Mcl-1, and GAPDH were purchased from Santa Cruz (Santa Cruz, CA, USA). Antibodies against CytC, Fas, FasL, and TRAIL were purchased from Abcam (Cambridge, MA, USA). Mouse anti-γ-H2AX antibody was from Millipore (Billerica, MA, USA). DMEM medium and FBS were purchased from Gibco BRL (Grand Island, NY, USA). FITC annexin V Apopotosis Detection Kit was from BD Biosciences (San Jose, CA, USA). Caspase-Glo 3/7, 8, and 9 assay kits and Plasmid extraction kit were purchased from Promega (Madison, WI, USA). N-acetyl-L-cysteine (NAC) and Reactive Oxygen Species Assay Kit were purchased from Sigma-Aldrich (St. Louis, MO, USA). Lipofectamine^TM^ 2000, Dichloro-dihydro-fluorescein diacetate (DCFH-DA), MitoSOX^TM^ Red, MitoTracker Deep Red FM, MitoTracker Green FM and Hoechst 33342 were from Invitrogen (Carlsbad, CA, USA). Trypan Blue Staining Cell Viability Assay Kit was from Beyotime Institute of Biotechnology (Jiangsu, China). Tetramethylrhodamine methyl ester (TMRM) was from Life Technologies (Carlsbad, CA, USA). Z-LEHD-FMK, Z-IETD-FMK, Z-DEVD-FMK were from Calbiochem (San Diego, CA, USA). Colorimetric TUNEL Apoptosis Assay Kit was from Beyotime (Shanghai, China). Mitochondria/Cytosol Fractionation Kit was from BioVision (San Francisco, CA, USA). DL-2000 DNA Marker was from TaKaRa Biotechnology (Dalian, China).

### 2.3. Preparation and Staining of Pig Placenta Tissue Sections

The placenta tissues were fixed with 10% neutral buffered formalin (Sigma), routinely processed, and embedded in paraffin. Series sections were cut at 4 μm thickness, and stained with hematoxylin and eosin (HE). The tissue damage was evaluated under light microscope. 

### 2.4. Detection of Apoptosis in Placenta Tissues

The placenta tissue sections prepared above were stained with TUNEL apoptosis kit for detecting the tissue apoptosis according to the manufacturer’s instructions. 

### 2.5. Construction of PPV NS1, VP1, and VP2 Expression Vectors

To construct PPV-encoded protein gene expression vectors, the NS1, VP1, and VP2 genes were amplified from PPV genome by PCR using the specific primers (Table 1).

Heat recycling conditions: initial denaturation at 94 ℃ for 5 min; followed by 30 cycles of denaturation at 94 ℃ for 1 min, annealing at different temperature (58 ℃, 56 ℃, and 57 ℃, respectively) for 1 min; extension at 72 ℃ for 4 min; followed by final extension at 72 ℃ for 10 min. The amplified PCR products were purified using the TIANgel Midi DNA purification kit. The three genes, without stop codons, were separately cloned into pcDNA3.1A plasmid to generate His-tag-fused NS1, VP1, and VP2 expression vectors. The expression vectors were confirmed by restriction enzyme digestion and sequencing.

### 2.6. Infection and Transfection

For infection by PPV, PK-15 cells were cultured in 6-well plates in DMEM complete medium at a density of 2.5 × 10^6^ cells/mL and infected with PPV at a multiplicity of infection (MOI) of 2. For PK-15 cell transfection, PK-15 cells were cultured in 6-well plates and transfected with the expression vectors using Lipofectamine 2000 following the manufacturer’s protocol when the cells reach 90% confluence. The pcDNA3.1A vector was transfected under the same conditions as a negative control.

### 2.7. Cell Viability Assay

PK-15 cell viability was evaluated using the Trypan Blue Staining Cell Viability Assay Kit (Beyotime). The cells in T75 flask were harvested and resuspended in 100 μL cell suspension solution (2.5 × 10^6^ cells/mL) and mixed with equal volumes of trypan blue solution for 3 min. Cell number was counted and viability was determined by the software CountStar Medical from Ruiyu (Shanghai, China).

### 2.8. Mitochondria and Cytosol Fractionation

The fractionation of mitochondria and cytosol was performed with Mitochondria/Cytosol Fractionation Kit from BioVision (San Francisco, CA, USA) ccoding to the manufacturer’s instruction.

### 2.9. Western Blot

The intrinsic apoptosis related proteins (Bax, P21, P53, Bcl-2, and Mcl-1) and recombinant PPV proteins (NS1, VP1, and VP2) were detected by Western blot. Following treatment, PK-15 cells cultured in T25 flasks were harvested at designated time and lysed with lysis buffer (5 mM Tris-HCl, 25 mM KCl, 2 mM EGTA, 2 mM EDTA, 1% NP-40, 15 mM NaCl, and protease inhibitors). The lysed samples were separated by 10 or 12% SDS-PAGE, electroblotted onto nitrocellulose, and incubated separately with primary antibodies against porcine Bax, P21, P53, Bcl-2, Mcl-1, or His-tag, followed by alkaline phosphatase-conjugated secondary antibody, and visualized by staining with nitro-blue tetrazolium and 5-bromo-4-chloro-3’-indolyphosphate (NBT/BCIP) (Bio-Rad). Western blot for CytC in the cytosol was performed using cytosol fraction.

### 2.10. Annexin V/PI Assay

The annexin V/PI assay was performed using the FITC Annexin V Apopotosis Detection Kit I (BD Biosciences), following the manufacturer’s instructions. Briefly, treated PK-15 cells were collected with a plastic scraper, washed with cold PBS twice, and resuspended in binding buffer at a concentration of 1 × 10^6^ cells/mL. One-hundred microliter aliquots of cell suspension were transferred into 1.5-mL tubes. Five microliters of both annexin V-fluorescein isothiocyanate (FITC) and propidium iodide (PI) were added, mixed by gentle tapping, and incubated at room temperature (25 ℃) in the dark for 15 min. Following addition of 400 μL of binding buffer to each tube, the cell suspensions were analyzed by flow cytometry (FACSCaliber, BD Biosciences, San Jose, CA, USA). Cell apoptosis was analyzed using CellQuest software (BD Biosciences). 

### 2.11. Detection of Caspase-3, -8, and -9 Activities

Caspase-3, -8, and -9 activities in treated PK-15 cells were determined using Caspase-Glo-3/7, -8, and -9 assay kits (Promega), following the manufacturer’s instruction.

### 2.12. Intracellular ROS and Mitochondrial ROS Detections

Intracellular ROS in PK-15 cells was detected with 2,7-dichlorofluorescein diacetate (DCFH-DA), which is oxidized in the presence of ROS and converted into highly fluorescent DCF [16]. PK-15 cells in 6-well plates were washed with PBS and incubated with 10 μmol/L DCFH-DA in the dark at 37 ℃ for 30 min, washed with PBS, and then incubated with 2 μg/mL Hoechst 33342 for nuclear staining. After washing with PBS, ROS levels were directly determined by fluorescence microscopy and quantitatively analyzed by flow cytometry.

Mitochondria-derived ROS was measured by flow cytometry using MitoSOX^TM^ (Life Technologies) as previously described [17]. MitoSOX^TM^, a dye that stains superoxide specifically generated from mitochondria, was used to quantify the relative amounts of mitochondrial ROS after stimulation. Briefly, PK-15 cells in 6-well plates were infected with PPV (MOI = 2) or transfected with NS1 vector (4 μg). Then 2.5 µmol/L of MitoSOX were loaded at various times. The cells were then washed 4 times with PBS and levels of mitochondrial ROS were measured in a FACSCaliber flow cytometer (BD Biosciences, San Jose, CA, USA), analyzed using FlowJo software (Treestar, OR, USA), and presented by MitoSOX relative fluorescence.

### 2.13. Mitochondrial Damage Measurement

Mitochondrial damage was examined by flow cytometry using MitoTracker Deep Red (Invitrogen) and MitoTracker Green (Invitrogen). MitoTracker Deep Red is a red fluorescent dye that stains mitochondria of living cells; MitoTracker Green is a green fluorescent dye that stains mitochondria of all cells. PK-15 cells cultured in 6-well plates were infected with PPV (MOI = 2) and transfected with NS1 vector (4 μg), then stained together with 0.5 μM MitoTracker Deep Red and 0.2 µM MitoTracker Green for 30 min at 37 ℃ in the dark. After incubation, cells were washed twice gently with PBS to remove excess unbound dye and reincubated with fresh PBS. Damage to mitochondria was measured in a FACSCaliber flow cytometer and analyzed using CellQuest software from BD Biosciences (San Jose, CA, USA).

### 2.14. Mitochondrial Membrane Potential (Δψm) Measurement

Mitochondrial membrane potential (ΔΨm) was determined using TMRM (Life Techonlogies, Carlsbad, USA), a cell-permeant, cationic, and fluorescent red-orange dye, which readily enters the cell and is then sequestered by active mitochondria (with normal Ψm). The fluorescence signal is therefore very weak. During cell apoptosis, the ΔΨm is disturbed due to mitochondrial damage, the TMRM cannot be sequestered, and fluorescence can be detected. The procedure: PK-15 cells were trypsinized, resuspended in 100 μL 200 nM TMRM in a 1.5 mL tube at a cell density of 5 × 10^6^ cells/mL, and incubated at 37 ℃ for 30 min before staining. After washing with washing buffer (PBS/ 50 nM TMRM), the cells were resuspended in 340 μL PBS and 100 μL aliquots of each sample in triplicate were added to wells of a 96-well plate. The fluorescent intensities were measured by spectrofluorophotometry (excitation 540 nm, emission 574 nm).

### 2.15. DNA Damage Detection

DNA double-strand breaks (DSBs) were used to evaluate DNA damage by detecting phospho-histone 2AX (γ-H2AX), a marker for DSBs [18]. PK-15 cells in 6-well plates were collected at various time points and lysed with Radioimmunoprecipitation assay (RIPA) buffer (10 mM Tris-Cl (pH 8.0), 1 mM EDTA, 0.5 mM EGTA, 1% Triton X-100, 0.1% sodium deoxycholate, 0.1% SDS, 140 mM NaCl, and 1 mM PMSF) and centrifuged at 12,000× *g* at 4 ℃ for 30 min. The γ-H2AX in the supernatant sample was detected by Western blotting using mouse anti-γ-H2AX antibody (Millipore) and alkaline phosphatase-conjugated goat anti-mouse IgG developed with NBT/BCIP (Bio-Rad).

### 2.16. Cell Cycle Analysis

The cell cycle was analyzed by measuring cell cycle phases by flow cytometry as previously described [19]. PPV-infected (MOI = 2) and NS1 gene (4 μg) transfected PK-15 cells in 6-well plates were trypsinized, washed with PBS, and fixed with 75% ethanol overnight at −20 ℃. The fixed cells were washed with PBS and stained with propidium iodide (PI) solution (PBS containing 30 µg/mL PI and 0.1 mg/mL RNase, pH 8.0) for 3 h in the dark. The stained cells were analyzed by flow cytometry. The cell cycle distribution was analyzed using MultiCycle software (Phoenix Flow Systems, San Diego, CA, USA). 

### 2.17. Statistical Analysis

The significance of differences between groups was evaluated by one-way analysis of variance (ANOVA) with Dunnett’s postcomparison test for multiple groups to control group, or by Student’s *t* test for two groups. All the presented data are shown as mean ± SD. * *p* < 0.05, ** *p* < 0.01. NS, no significance.

## 3. Results

### 3.1. PPV Infection Induces Placenta Tissue Apoptosis and Damage in the Pregnant Sows

To investigate whether PPV infection can cause placenta tissue apoptosis and damage, the placenta tissues were collected from normal labored and aborted sows from the local pig farms, and only PPV-induced (no other viruses) aborted placenta tissues and normal labored placenta tissues were histologically sectioned, the histopathological changes were analyzed by hematoxylin-eosin staining, and the apoptosis was detected by TUNEL staining. Results showed that the significant damage in placental fold tissue (Figure 1B) was observed in PPV-induced aborted sows, not in normal labored sows (Figure 1A). PPV infection could increase the TUNEL signal in the placental fold tissue (Figure 1D) compared with normal placenta tissue (Figure 1C), suggesting that PPV infection can cause DNA damage of the cells in the PPV-induced aborted placenta tissues, indicating that PPV can induce the apoptosis of the host cells. We also measured PPV titers in the PPV-induced aborted placenta tissues and showed that the high PPV titers were detected in the PPV-induced aborted placenta tissues. All these results indicate that PPV can induce the placenta tissue apoptosis and damage in the pregnant sows.

### 3.2. PPV-Induced Apoptosis of PK-15 Cells is Mainly Related to NS1 Protein

A previous report showed that PPV infection could induce apoptosis of PK-15 cells [14]. To investigate which PPV-encoded protein is involved in the apoptosic induction, the three PPV-encoded protein genes (NS1, VP1, and VP2) were amplified by PCR (Figure 2A), and their His-tag-fused expression vectors were constructed (Figure 2B). All vectors were used separately to transfect PK-15 cells to test whether the constructs could mediate their corresponding gene to be expressed in the host cells. As shown in Figure 2C, the specific protein bands were detected by Western blot using anti-His antibody, indicating that the constructed vectors in this study did mediate their gene expression in PK-15 cells. However, the expression levels of the three proteins mediated by their vectors were differed considerably (NS1 > VP1 > VP2). 

To determine which protein contributes to cell apoptosis, we first compared the expression of these three proteins under the transfection with different doses of their vectors in order to standardize the protein levels for further investigation on these three protein functions in cell viability and apoptosis. Results showed that the same expression levels for these three proteins were observed by transfecting about 1 μg NS1, 2μg VP1, and 4 μg VP2 vectors (Figure 2D). So we adopted the doses of vectors to compare the effects of the three proteins on cell viability and apoptosis in the condition of same expression levels. The freshly seeded PK-15 cells (2.5 × 10^6^ cells/mL) in 6-well plate were separately infected with PPV (MOI = 0.5, 1, 2, and 4) and transfected with the three vectors (1 μg NS1, 2μg VP1, and 4 μg VP2). The cell viability was detected at different time (12, 24, and 48 h) after PPV infection and at 48 h after vector transfections. The phosphatidylserine eversion and caspase-3 activity were detected at 48 h after PPV infection and the vector transfections. Results showed that both PPV infection and NS1 vector transfection significantly reduced the cell viability, as detected by the trypan blue dye exclusion test (Figure 2E,F), and increased phosphatidylserine eversion (Figure 2G,H,I) and caspase-3 activity (Figure 2J). However, like the empty vector-transfected cells, VP1 and VP2 vector-transfected cells did not significantly affect the above parameters during the test period (Figure 2H,J), indicating that NS1, but not VP1 or VP2, plays a crucial role in PPV-induced apoptosis. 

### 3.3. NS1 Induces Intrinsic Apoptosic Pathway in the PK-15 Cells

There are two major signal pathways for apoptotic induction: extrinsic and intrinsic. To determine which of these was induced by NS1, we analyzed the activities of caspase 8, 9, and 3 in PK-15 cells at 12 and 24 h after PPV infection and 24 and 48 h after NS1 gene vector transfection. Results showed that both PPV infection and NS1 vector transfection significantly increased caspase 9 and 3 activities in the cells (Figure 3A,B). However, caspase 8 activity did not change significantly (Figure 3A,B). These results suggest that NS1-mediated PPV-induced PK-15 cell apoptosis is affected mainly via the intrinsic signal pathway. To provide further evidence for this, we applied specific caspase inhibitors (Z-LEHD-FMK for caspase 9, Z-IETD-FMK for caspase 8, and Z-DEVD-FMK for caspase 3) so as to determine the relevant caspases in NS1-induced apoptosis. Data in Figure 3C show that NS1-induced caspase 3 activity was significantly inhibited by Z-LEHD-FMK and Z-DEVD-FMK, but not by Z-IETD-FMK, indicating that NS1-induced apoptosis mainly involves the caspase 9-mediated intrinsic apoptosic pathway. Figure 3D showed the cell survival rates under the different treatments. Moreover, intrinsic mitochondria-related proapoptotic molecules (Bax, P21, and P53), antiapoptotic molecules (Bcl-2 and Mcl-1), and extrinsic proapoptotic molecules (Fas, FasL and TRAIL) were detected in NS1 gene-transfected PK-15 cells by Western blot, revealing that NS1 could enhance expression of intrinsic proapoptotic molecules Bax, P21, and P53 (Figure 3E) and inhibit expression of intrinsic antiapoptotic molecules Bcl-2 and Mcl-1 (Figure 3F), causing cytochrome C release (Figure 3G,H). However, NS1 did not significantly affect expression of extrinsic apoptotic molecules (Fas, FasL, and TRAIL) (Figure 3I). Figure 3J shows NS1 expression levels at different transfection times. Altogether, it can be concluded that NS1-induced apoptosis mainly involves the intrinsic, not extrinsic, apoptotic pathway.

### 3.4. NS1 Induce Intracellular and Mitochondrial ROS Accumulations

Reactive oxygen species (ROS) accumulation is considered as a crucial inducer of intrinsic apoptotic induction. Previous work has indicated that PPV infection can induce ROS accumulation, thereby inducing apoptosis in host cells [15]. However, whether NS1 can induce ROS accumulation remains unknown. Hence, PK-15 cells were infected with PPV (MOI = 2) or transfected with NS1 vector (4 μg) separately, and intracellular ROS and mitochondrial ROS were analyzed by DCFH-DA fluorescence assay and MitoSOX staining/flow cytometry, respectively, at different time points. Results showed that both PPV and NS1 significantly promoted accumulation of intracellular ROS (Figure 4A,B) and mitochondrial ROS (Figure 4C,D), indicating that NS1 protein contributed mainly to PPV-induced ROS accumulation in PK-15 cells.

### 3.5. ROS Accumulation Is Mainly Involved in NS1-Induced Apoptosis

To determine whether NS1-induced mitochondrial ROS plays a dominant role in PPV- and NS1-induced apoptosis, the PK-15 cells were pretreated with the ROS inhibitor N-acetyl-L-cysteine (NAC, 2 mM) for 2 h, then infected with PPV (MOI = 2) or transfected with NS1 vector (4 μg). At 24 h after PPV-infection and 48 h after NS1 gene transfection, phosphatidylserine eversion, caspase 3 activity, and cell viability were determined. Results showed that, in the presence of NAC, the annexin V/PI positive cells and caspase 3 activity were significantly decreased to levels seen without PPV infection or NS1 transfection (Figure 5). These results clearly demonstrate that NS1-induced ROS accumulation is involved in the activation of PK-15 cell apoptosis.

### 3.6. PPV- and NS1-Induced ROS Accumulation Induces DNA Damage Leading to Cell Cycle Arrest at the G1 and G2 Phases

NS1 was identified within the nucleus, in which it is not only involved in viral replication and gene transcription, but also in viral host cell pathogenicity [20]. Whether PPV NS1 protein in the nucleus can induce host DNA damage, and whether NS1-induced ROS plays a crucial role in NS1-induced DNA damage, was unclear. To address these issues, we analyzed DNA damage by checking DNA double-strand breaks (DSBs), the most severe form of DNA damage, in PPV-infected and NS1 vector-transfected PK-15 cells, determined by measuring the appearance of γ-H2AX, which results from the phosphorylation of histone H2AX at Ser 139 and presents a well-established indicator of the formation of DSBs [21]. As shown in Figure 6A,B, a significant increase in γ-H2AX was detected at 24 h after PPV infection and 48 h after NS1 vector transfection, indicating the presence of DSBs in PPV-infected and NS1 vector-transfected cells. The ROS inhibitor NAC significantly inhibited increase of γ-H2AX, i.e., it inhibited PPV- and NS1-mediated induction of DSBs, suggesting that ROS accumulation might contribute to PPV- and NS1-induced DNA damage, and consequently affecting DNA replication and cell differentiation as shown in Figure 6C. Cell cycle analysis showed that upon PPV infection or NS1 vector transfection, the cell populations in G1 and G2 phases were increased, indicating that both PPV and NS1 have the ability to arrest cells in G1 and G2 phases, which will promote the apoptosis of the host cells. Interestingly, NS1 vector-induced cell cycle arrest in G2 phase was stronger than that induced by PPV (Figure 6C). It can be also seen that ROS inhibitor (NAC) can partially reverse the PPV- and NS1-induced cell cycle arrest in G1 and G2 phases, indicating that ROS plays an important role in this process. 

### 3.7. PPV- and NS1-Induced ROS Accumulation Causes Mitochondrial Damage

Mitochondria play an important role in cells since they not only produce energy needed for the cell biological activities (ATP) [22], but also regulate redox balance, apoptosis [23], autophagy [24], immunity, and inflammation [25,26,27,28,29]. Mitochondrial damage will certainly affect their regulatory functions. Mitochondrial damage and dysfunction are considered important factors in mitochondria-mediated apoptotic induction. To determine whether PPV- and NS1-induced ROS accumulation can cause mitochondrial damage and dysfunction, we used PPV and the NS1 gene vector to infect and transfect PK-15 cells, respectively. PPV- and NS1-induced mitochondria damage in living PK-15 cells were analyzed at 24 h post-PPV infection and 48 h post-NS1 vector transfection by fluorescent dye staining/flow cytometry using mitochondria-specific fluorescent dyes (MitoTracker Deep Red FM and MitoTracker Green FM) and by measurement of ΔΨm loss. Results showed that both PPV and NS1 significantly decreased the proportion of MitoTracker Deep Red FM staining-positive cells (healthy cells) in a dose-dependent manner (Figure 7A), indicating that both PPV and NS1 can induce mitochondrial damage. Furthermore, the ROS inhibitor NAC significantly inhibited PPV- and NS1-induced mitochondrial damage, suggesting that the mitochondrial damage is mainly caused by PPV- and NS1-induced ROS accumulation. The same result was observed in the PPV- and NS1-induced loss of mitochondrial membrane potential (ΔΨm) detected by TMRM staining/flow cytometry (Figure 7B,C), showing that both PPV and NS1 could induce ΔΨm loss in a dose-dependent manner. Also, NAC significantly reduced theΔΨm loss, further indicating that ROS accumulation is involved in PPV- and NS1-induced mitochondrial damage. It can be seen from above results that the ROS accumulation plays an important role in PPV- and NS1-induced mitochondrial damage.

## 4. Discussion

Virus-mediated apoptosis plays a crucial role in viral pathogenesis. Parvoviruses from different animals have been identified to have the ability to induce host cell apoptosis, resulting in host cell and target tissue damage. Parvoviruses preferentially infect rapidly dividing cells, such as bone marrow progenitors, intestinal epithelial cells, and placental cells, can induce cell apoptosis and death and interfere with cell renewal resulting in cellular dysfunction. For example, human parvovirus B-19 can infect human megakaryocyte–erythroid progenitor cells causing bone marrow disorder and chronic aplastic anemia. Once a pregnant woman is infected by the B-19 parvovirus, the virus commonly attacks the placenta and fetus causing abortion [30]. Canine parvovirus mainly infects proliferating crypt epithelial cells in the intestinal mucosa of puppies, causing intestinal mucosal epithelial cell damage and destroying the natural barrier of the intestinal tract. The resulting secondary infections lead to acute hemorrhagic inflammation and death [31]. Porcine parvovirus is one of the most important infectious diseases that cause reproductive failure in pigs. PPV infection of pregnant pigs can result in abortion, stillbirth, malformed fetus and mummification. Studies have shown that PPV can infect the dividing cells in the placenta and fetus tissues, resulting in a series of symptoms of reproductive failure.

To investigate the pathogenesis of PPV, by consulting the experimental results from other parvoviruses, several groups [11,32,33] have investigated PPV effects on cell apoptosis and showed that PPV can induce host cell apoptosis [34], which appeared to be related to PPV-induced ROS accumulation [15]. However, the molecular mechanism underlying PPV-induced apoptosis remained largely unknown. Many questions had not been answered. For example, which protein encoded by PPV is involved in apoptosis induction and ROS accumulation? What changes result from the accumulation of ROS? What is known about the intrinsic relations between ROS accumulation and PPV-induced DNA and mitochondrial damage? To address these questions, in this study, we first constructed expression vectors for PPV-encoded NS1, VP1, and VP2 genes and analyzed their capacities to induce cell apoptosis, and showed that NS1 protein, but not VP1 or VP2 proteins, plays a major role in PPV-induced apoptosis, which is consistent with our previous studies on the apoptosis caused by NS1 protein of canine parvovirus [35].

However, it is worth noting that parvovirus NS1 gene contains full-length NS2 gene, so NS1 gene expression vector should also express NS2 protein, which may have potential influence on PPV-induced apoptosis. Therefore, the effect of NS2 on PPV-induced apoptosis cannot be excluded in this study. Therefore, in future experiments, NS1 expression vector expressing only NS1 but not NS2 should be constructed by mutating the intron splicing site of NS1 gene without changing the amino acid sequence of NS1 protein, so as to further confirm the effect of NS1 on PPV-induced apoptosis. 

Our data showed that PPV NS1 protein can induce apoptosis mainly via the endogenous mitochondrial pathway. PPV- and NS1-induced ROS accumulation is a crucial factor for their induction of apoptosis. ROS accumulation was identified to be a crucial factor causing DNA and mitochondrial damages in the host cells. Cell cycle arrest caused by DNA damage and induction of cytochrome C release and proapoptotic molecules (Bax, P21, and P53) play an important role in NS1-induced cell apoptosis. It can be seen from our results that PPV- and NS1-induced ROS accumulation leading to DNA and mitochondria damage is an important pathway for PPV-induced apoptosis. Therefore, the mechanism underlying PPV-induced reproductive failure can be proposed as shown in Figure 8.

It can be seen from Figure 8 that infection with PPV first induces ROS accumulation in the host cells through its encoded NS1 protein (possibly including NS2 protein) resulting in host DNA and mitochondrial damage, which in turn result in cell cycle arrest and expression of mitochondrial proapoptotic molecules and cytochrome C release, all of which together lead to host cell apoptosis. Finally, the PPV-induced apoptosis causes placental tissue damage and reproductive failure of the sows. Whether NS1 directly induces DNA and mitochondrial damage remains to be further investigated. It is undoubtedly of great significance to elucidate the mechanism of PPV-induced apoptosis for exploring the mechanism of parvovirus induced reproductive failure.

## 5. Conclusions

NS1 protein is the main contributor for PPV-induced host cell apoptosis, mainly involving NS1-induced host DNA damage, ROS generation, and mitochondrial damage. Our results provide an insight into the molecular basis for elucidation of PPV-induced host cell apoptosis and reproductive failure.

## Figures and Tables

**Figure 1 viruses-11-00389-f001:**
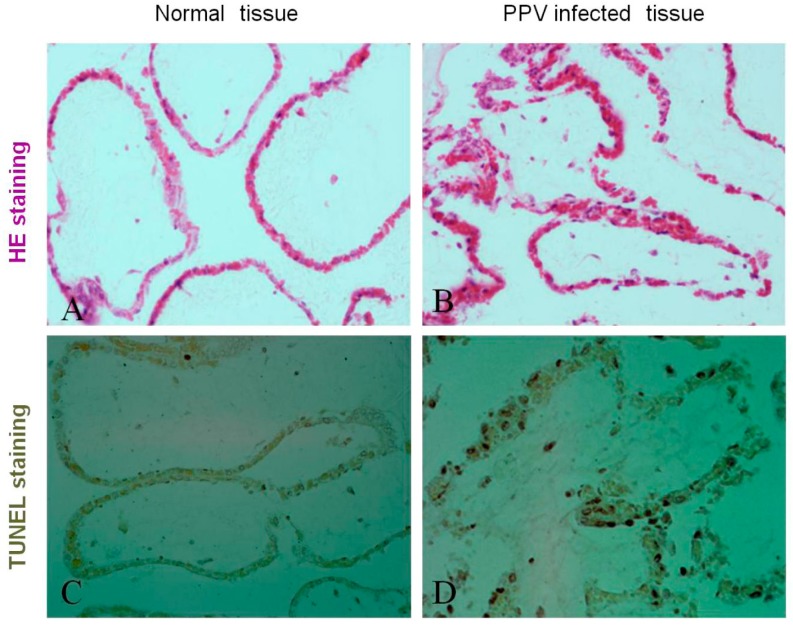
Porcine parvovirus (PPV)-induced placental tissue apoptosis and damage (400×). HE: Haematoxylin/eosin; TUNEL: transferase-mediated deoxyuridine triphosphate-biotin nick end labeling.

**Figure 2 viruses-11-00389-f002:**
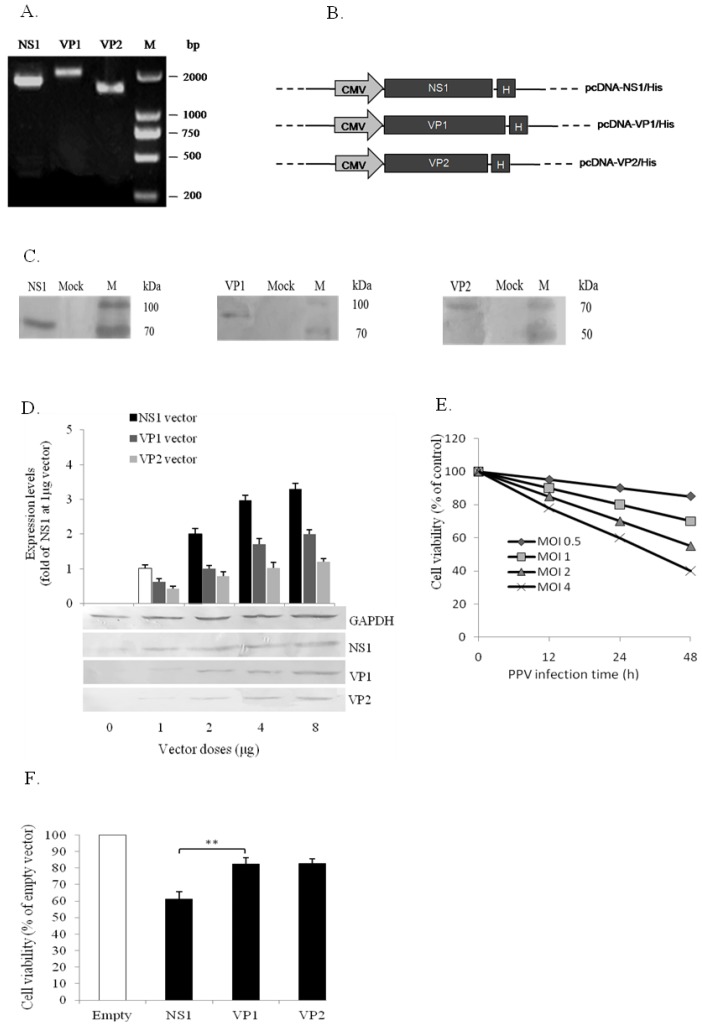
PPV-encoded protein expression and PPV protein- and PPV protein-induced apoptosis in PK-15 cells. (**A**) PCR products of PPV-encoded NS1 (1989 bp), VP1 (2162 bp), and VP2 (1740 bp) genes amplified from the PPV genome. M, DL-2000 DNA marker. (**B**) Constructs of eukaryotic expression vectors of PPV-encoded protein genes. All three PPV-encoded genes (NS1, VP1, and VP2) fused with His-tag were driven by the CMV promoter. (**C**) Expression of the PPV-encoded proteins in PK-15 cells using Lipofectamine 2000 transfection reagent, detected by immunoblotting using anti-His antibody. (**D**) Comparison of protein expression levels among NS1, VP1, and VP2. (**E**,**F**) show the cytotoxic effects of PPV infection and PPV-encoded protein gene transfections (1 μg NS1, 2μg VP1 and 4 μg VP2) on PK-15 cells, respectively. (**G**,**H**) show PPV- and PPV-encoded protein-induced phosphatidylserine eversion detected by annexin V-FITC/PI staining and flow cytometry and presented as a percentage of annexin V-FITC^+^/PI^+^ cells in total cells, respectively. (**I**) Induction by different doses of NS1 vector on phosphatidylserine eversion determined by the same method as above. (**J**) Caspase 3 activity in PK-15 cells after PPV infection and PPV-encoded protein gene transfection: empty, pcDNA3.1A vector transfection as a negative control. Data are presented as mean ± SD. * *p* < 0.05, ** *p* < 0.01.

**Figure 3 viruses-11-00389-f003:**
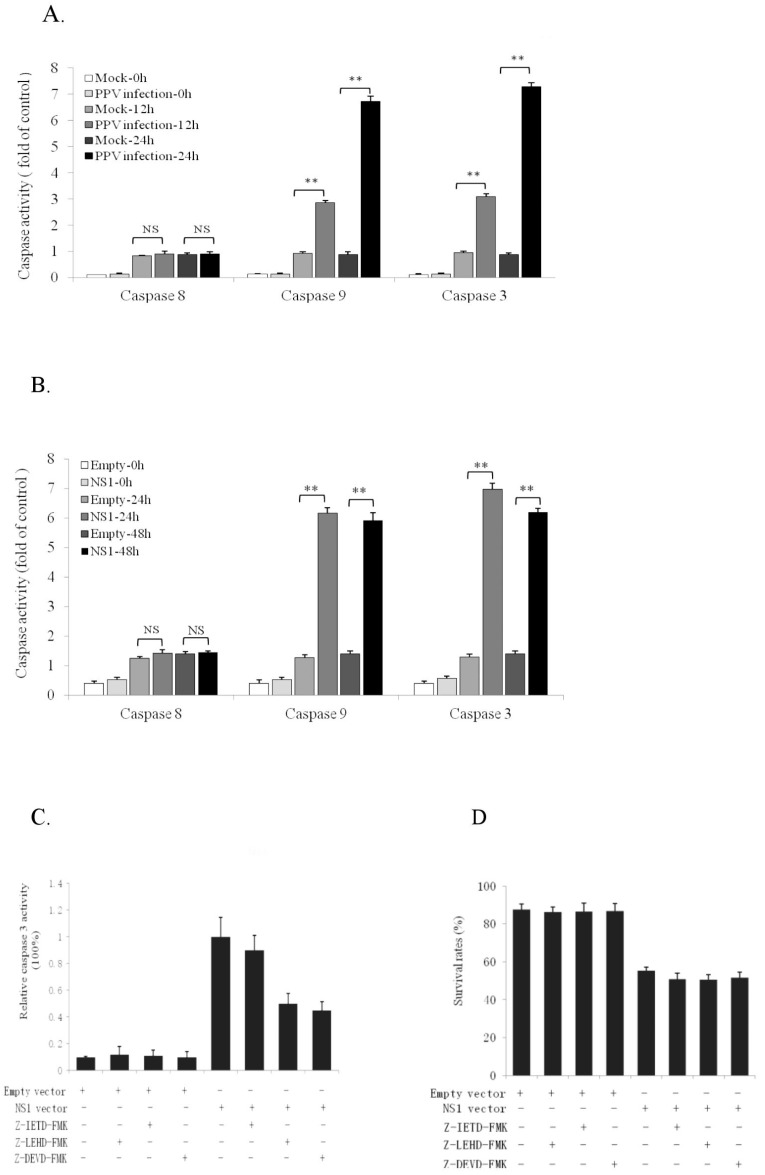
Effects of PPV, NS1, and inhibitors on caspase activities, and effects of NS1 on expression of pro- and antiapoptotic molecules. (**A**) Effects of PPV infection on the activities of caspases 8, 9, and 3 in PK-15 cells at different times. (**B**) Effects of NS1 gene vector transfection on caspase 8, 9, and 3 activities. (**C**) Inhibitor effects on caspase 3 activities upon NS1 gene transfection in the PK-15 cells. (**D**) Cell survival rates under the NS1 vector transfection in the different treatments with different caspase inhibitors. (**E**) Effects of NS1 gene transfection on Bax, P21 and P53 expression detected by immunoblotting. (**F**) Effects of NS1 gene transfection on Bcl-2 and Mcl-1 expression.(**G**,**H**) show the effects of PPV and NS1 genes on cyt C release from mitochondria, respectively. (**I**) Effects of NS1 gene transfection on Fas, FasL, and TRAIL expression. (**J**) NS1 expression levels at the different transfection time. Data are presented as mean ± SD. * *p* < 0.05, ** *p* < 0.01. NS, no significance.

**Figure 4 viruses-11-00389-f004:**
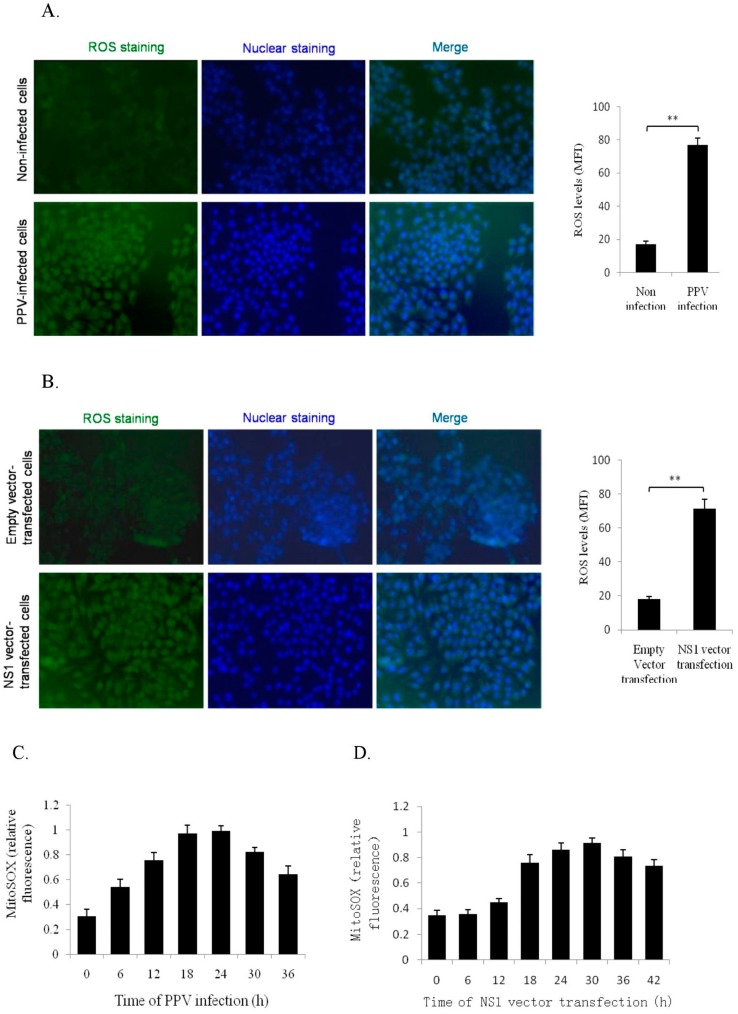
Effects of PPV and NS1 on mitochondrial ROS production. (**A**,**B**) show PPV- and NS1-induced ROS generation, respectively, determined using the ROS fluorescence detector DCFH-DA and an inverted fluorescence microscope (left, 200×), quantified by flow cytometry, and presented as mean fluorescent intensity (MFI) (right). (**C**,**D**) show the effects of PPV and NS1 on mitochondrial ROS at the different time points, respectively, detected by MitoSOX staining/flow cytometry and represented as MitoSOX relative fluorescence. Data are presented as mean ± SD. * *p* < 0.05, ** *p* < 0.01.

**Figure 5 viruses-11-00389-f005:**
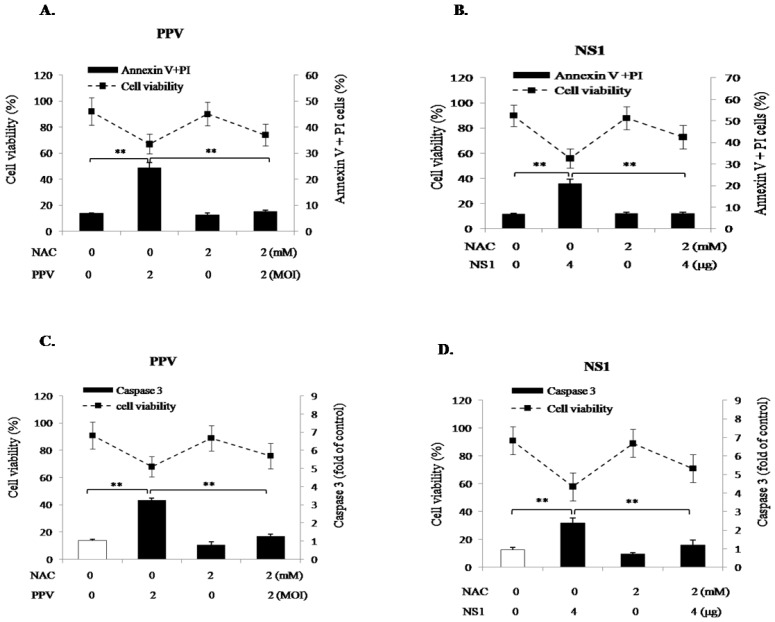
Effects of ROS inhibitor (NAC) on PPV- and NS1-induced cell viability and apoptosis. (**A**,**B**) show the effects of NAC on PPV- and NS1-induced phosphatidylserine eversion, respectively, presented as the percentage of annexin V/PI positive cells. (**C**,**D**) show the effect of NAC on PPV- and NS1-induced caspase 3 activation, respectively. Data are presented as mean ± SD. * *p* < 0.05, ** *p* < 0.01. NS, no significance.

**Figure 6 viruses-11-00389-f006:**
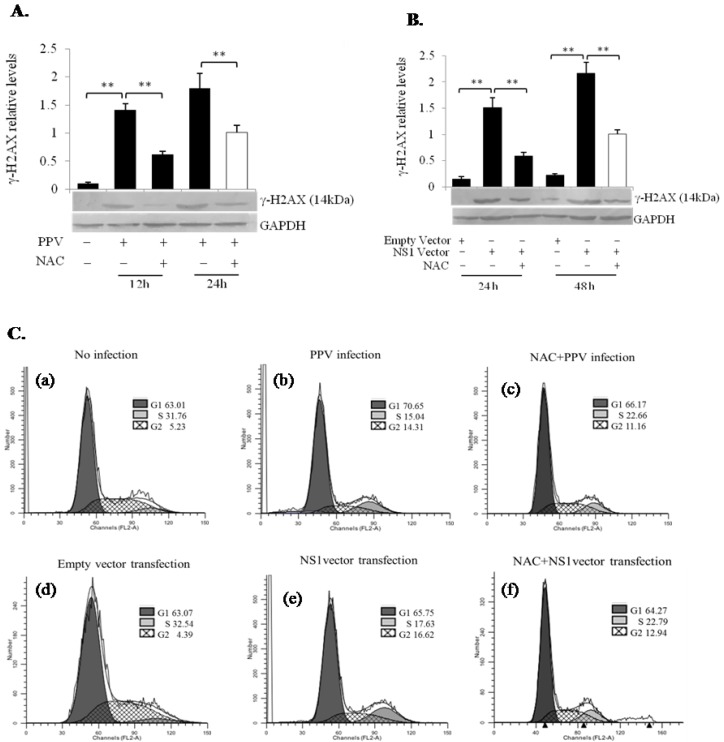
Effect of PPV and NS1 on γ-H2AX expression and the PK-15 cell cycle. (**A**) Effect of PPV infection on γ-H2AX expression at 12 and 24 h after infection. NAC, ROS inhibitor. (**B**) Effect of NS1 vector transfection on γ-H2AX expression at 24 and 48 h after NS1 vector transfection. Empty vector, pcDNA3.1A vector as a negative control. (**C**) Effect of PPV infection and NS1 transfection on cell cycle of the PK-15 cells (**a**,**b**,**d**,**e**) and effects of NAC on PPV- and NS1 vector-induced cell cycle (**c**,**f**). Data are presented as mean ± SD. * *p* < 0.05, ** *p* < 0.01.

**Figure 7 viruses-11-00389-f007:**
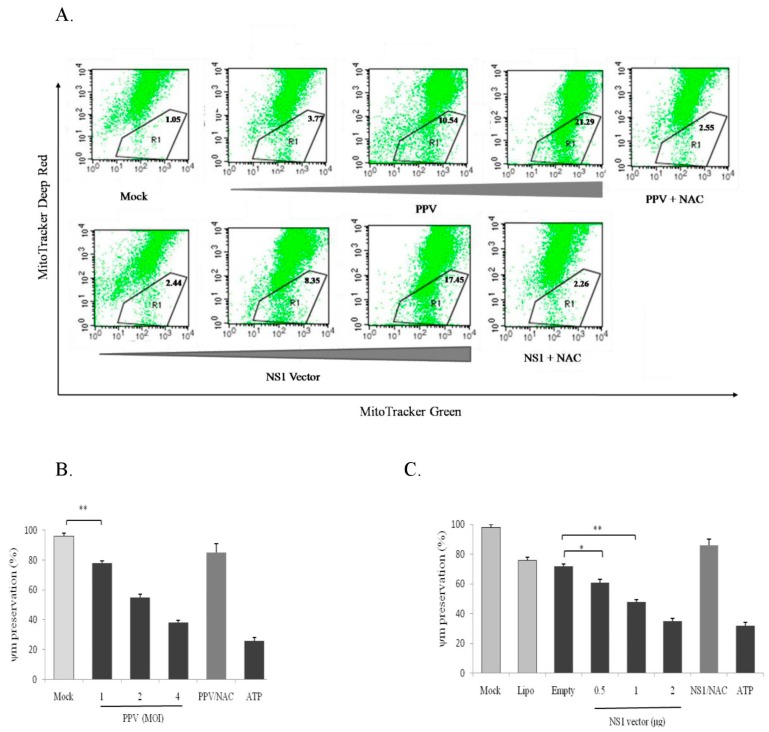
PPV- and NS1-induced mitochondrial damage and membrane potential (ΔΨm) loss. (**A**) Effects of PPV and NS1 on mitochondrial damage: MitoTracker Deep Red for health mitochondria staining and MitoTracker Green FM for total mitochondria staining. (**B**,**C**) effects of PPV and NS1 on mitochondrial membrane potential, respectively: ATP as a positive control for PPV and NS1-induced the loss of mitochondrial membrane potential. Data are presented as mean ± SD. * *p* < 0.05, ** *p* < 0.01.

**Figure 8 viruses-11-00389-f008:**
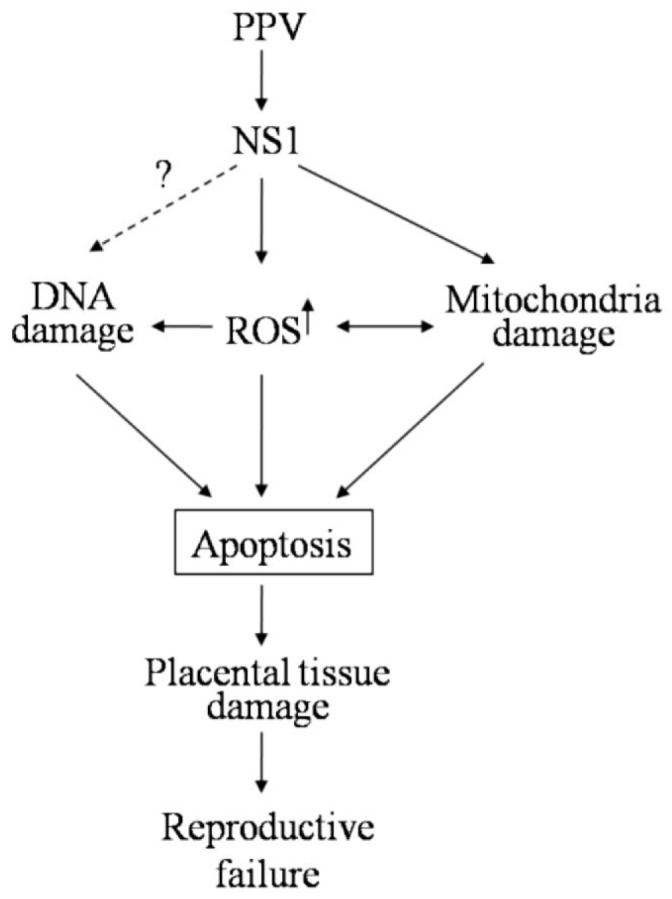
Mechanism of PPV-induced apoptosis resulting in reproductive failure of sows.

**Table 1 viruses-11-00389-t001:** Primers for amplifying porcine parvovirus (PPV)-encoded protein genes by PCR.

Gene	Primer Name	Primer Sequence (5′→3′)	Size/bp	Restriction Site
NS1	NS1-F	CGG*GGTACC*ACCATGGCAGCGGGAAACACTTAC	2007	*Kpn* I
	NS1-Rns	CCG*ACCGGT*TTCAAGGTTTGTTGTGGGTGC		*Age* I
VP1	VP1-F	CGG*GGTACC*ACCATGGCGCCTCCTGCAAAAAGAGCA	2179	*Kpn* I
	VP1-Rns	CCG*ACCGGT*GTATAATTTTCTTGGTATAAGTTG		*Age* I
VP2	VP2-F	CGG*GGTACC*ACCATGAGTGAAAATGTGGAACAAC	1758	*Kpn* I
	VP2-Rns	CCG*ACCGGT*GTATAATTTTCTTGGTATAAGTTG		*Age* I

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
