# Peer review of "Porcine Parvovirus Infection Causes Pig Placenta Tissue Damage Involving Nonstructural Protein 1 (NS1)-Induced Intrinsic ROS/Mitochondria-Mediated Apoptosis"

_viruses, 2019, doi:10.3390/v11040389_

Round 1
Reviewer 1 Report
The authors extend previous work showing that PPV infection leads to cellular apoptosis. They show that the PPV induced apoptosis is associated with the induction of ROS, DNA damage, and cell cycle arrest, and that these effects can be accomplished by the individual expression of the gene for the viral large nonstructural protein NS1. The work adds to similar studies on related parvoviruses CPV and H1. While there are some problems in wording and grammar, the manuscript is generally well written and well put together.
There are a number of issues that must be dealt with:
1. Fig. 2: Does the NS1 expression vector also encode the viral NS2 protein? If so constructs expressing NS1 alone, and NS2 alone should be examined.
To more clearly interpret the results in Fig. 2 (as well as other experiments in the manuscript), it is essential to have some quantification of the number of cells infected (2D and F) and successfully transfected (2E and F-I) in these experiments. Also, comparison of protein levels in the transfected cells (2F-I) is necessary.
2. Fig. 3: In panel 3C, evaluation of cell survival should be included. For panels D, E, G & H, a non-NS1 protein expression (negative) control should be added. Some indication of the expression levels of NS1 should be included (as well as a dose response?), and as above, some indication of the number of transfection-positive and infection-positive cells needs to be included.
3. Fig 4: For 4A and 4B, please include a nuclear stain. For Fig. 4D, is there a disconnect between the timing here, compared to the difference in timing seen between infection and transfection in Fig. 3B vs Fig. 3A? Please address.
4. Fig. 5: The labeling under Fig. 5A should be switched. Has the concentration of NAC been optimized for PK-15 cells? It is important to also report changes in cell viability following treatment.
5. Fig 6C: the data seem to indicate a G2 block, but much less strongly a block in G1. It is important to determine if the block can be reversed with the ROS inhibitor.
Author Response
see the uploaded

Reviewer 2 Report
This paper describes an original study in which the authors provide a molecular basis for elucidating PPV-induced cell apoptosis and reproductive failure. The authors demonstrate that NS1 protein is the main contributor for PPV- induced host cell apoptosis via NS1-induced host DNA damage, ROS generation and mitochondrial damage. PPV NS1 can induce host cell apoptosis and death, so playing the crucial role in PPV-induced placental tissue damages.
Despite the complexity the paper is well written and organized. The methodological part is described in detail, including used reagents.
Minor comments:
Line 238 - M, DL-2000 DNA marker - there is no indication of who the manufacturer is or from where it was purchased;
Line 245 – "Induction by different 4 of NS1 vector ...." is not understandable, should be corrected;
Lines 253-254 - What the authors have meant by “However, neither significantly changed caspase 8 activity although there was a tendency towards this following PPV infection”, in particular by “tendency towards this”? Should it be seen in the Figure 3A? But it's not visible.
Line 290 - Should be (A) and (B);
Line 293 - Should be (C) and (D);
Fig. 2, 3, 5, 6 and 7 – it is desirable to add an explanation to the signature *P < 0.05, **P < 0.01 and NS - no significance, to the reader's comfort so that you do not have to return to the section of 2.16 Statistical analysis every time;
Page 13, Fig. 5 - size and sharpness of the picture could be increased, because the recordings on the axes are difficult to see.
Author Response
See the uploaded

Reviewer 3 Report
Zhang et al. describe experiments conducted to demonstrate that the non-structural protein 1 (NS1) of porcine parvovirus (PPV) is involved in virus-induced apoptosis of placental tissue and porcine kidney cells PK-15. The data suggest that NS-1-induced increase of ROS causes mitochondrial damage and DNA damage. The sequela is an increase of apoptosis of fetal and placental cells which may explain PPV-associated reproductive failure. The data presented in this manuscript support this concept. A few issues need to be improved: 1. line 40: Capitalize and italicise "parvovirus"; italicise "Parvoviridae". 2. line 45: "capsid proteins" should be replaced by "virus capsid". 3. line 50: It should read "PPV infection across the placental barrier..." 4. line 87: Carlsbad 5. lines 120-121: Sentence is unclear, please rephrase. 6. line 136: aliquots 7. line 245: "(H) Induction by different 4 on NS1..." Sentence is unclear, please rephrase. 8. line 290, legend to Figure 4: It should read "(A) and (B) PPV and NS1-induced..." 9. line 293, legend to Figure 4: It should read "(C) and (D) Effects of PPV and NS1 on..." 10. line 310: It should read "NS1 was identified localized within the nucleus..." 11. lines 319-325: The data presented in Fig. 6C demonstrate a moderate increase of cells that pause in G1 arrest after NS1 plasmid transfection compared to PPV infection. In contrast, a stronger increase in the number of cells that remain in G2 arrest is observed compared to the PPV-infected cells. This should be described in lines 319-325 more clearly. 12. Figure 1 is not convincing. The authors have to present better figures showing apoptotic cells by TUNEL assay. The TUNEL assay is not well-described. 13. Figures 3 F, G: The figure indicates that cytochrom C increases in a time-dependent manner after infection with PPV or transfection with NS1-plasmid. Why is a strong CytC-band in mock-infected and mock-transfected lanes, respectively? There should be no band as in non-infected cells cytochrom C is in mitochondria rather than in cytosol. 14. Usually, there is a time lag in the transfection experiments compared to the infection experiments (compare Figs. 3A/3B, 5, 6A/6B). However, in Figs. 3 and 4 there no time lag. I would expect that it takes some time after liposome transfection for a plasmid to reach the nucleus and induce mRNA transcription. 15. Caption of Figure 4A: replace "uninfection" by "non-infected cells"
Author Response
See the uploaded

Round 2
Reviewer 1 Report
The authors have made a number of useful changes; however, there are still some issues that need to be addressed.
The authors misunderstood my criticism of Fig. 2. If their NS expression vector is indeed expressing NS1 and NS2 then the results shown cannot be attributed to NS1 alone as suggested. (In addition, if the proteins are his-tagged on the C-terminus it should be mentioned in the Materials and Methods.) In addition, to compare the results in Fig. 2 E and F-I it is essential to standardize to protein expression levels following transfection - not just use equivalent amounts of plasmid. Finally, for 2D and F, some quantification of actual infection rates is necessary for comparative purposes (it certainly won’t be 100%) - not just reporting the MOI.
For Fig. 5, data showing the extent of the increase in cell viability should be shown.
For Fig. 6, it is important for the main point of the paper to demonstrate whether the block is reversed with the ROS inhibitor.
Author Response
See the PDF file

Reviewer 2 Report
The authors have taken into account all the objections and have made appropriate corrections.
Author Response
Thank you very much!
Reviewer 3 Report
The revised manuscript presents improved. Howver, there are still some typos and grammatical flaws that need minor revision.
line 128: manufacturer's instruction
line 157: staining
line 221-222: The modified sentence still needs improvement.
line 240: The cell viability was detected...
line 303: caspase 8 activity did not change significantly...
line 317: The modified sentence still needs improvement.
line 386: The modified sentence still needs improvement.
lines 471-476: The modified sentence still needs improvement.
Author Response
see the PDF

Round 3
Reviewer 1 Report
The manuscript is now fine, except, please add one sentence somewhere indicating that the NS expression vector also expresses NS2 and potential effects of NS2 cannot be ruled out.
Author Response
Please check the PDF file

Reviewer 3 Report
After the second revision, the manuscript of Zhang et al. presents further improved. The authors addressed most of the concern raised by the reviewers.
Further issues still need improvement:
1. Re: Comments by reviewer #1: Co-expression of NS1 and NS2.
I understand that it was not the authors' aim to investigate NS2 and that only NS1 is provided with a his-tag.
However, if NS2 is also expressed upon transfection of NS1 plasmid, then it might contribute to the results of the experiments. The authors should include a statement that they cannot exclude a possible contribution of the NS2 protein. It is not sufficient to state that NS2 has not been recognized yet as an important protein in parvovirus pathogenicity of human and canine parvovirus. The authors are advised to use a modified NS1 plasmid in their future experiments which allows expression of NS1 but abolishes NS2 expression (e.g. by introduction of a premature stop codon or prevention of alternative splicing).
2. Typos and the English style of the newly modified text passages need improvement. Examples:
lines 237-238: "However, the expression levels of the three proteins mediated by their vectors differed considerably (NS1 > VP1 > VP2)."
line 310: "Comparison...".
line 339: "Figure 3J shows NS1 expression levels at different transfection times."
line 405: "NS1 expression levels at different transfection times."
lines 452-454: "At 24 h after PPV-infection and 48 h after NS1 gene transfection, phosphatidylserine eversion, caspase 3 activity and cell viability were determined."
lines 495-496: "Interestingly, NS1 vector-induced cell cycle arrest in G2 phase was stronger than that induced by PPV (Figure 6C)."
lines 521-523: The sentence is not clear.
line 527: "Mitochondria play an important role in the cell..."
Author Response
Please check the PDF file
